SOFTWARE

# VesiclePy: A machine learning vesicle analysis toolbox for volume electron microscopy

**Jason Ken Adhinarta[1]☉, Yutian Fan[1]☉, Adam Gohain[1]☉, Michael Lin[1]☉, Paige Nurkin[2]☉, Richard Ren[2]☉, Micaela Roth[1]☉, Shulin Zhang[2]☉, Ayal Yakobe[2], Rafael Yuste[2], Donglai Wei[1]***

**1** Computer Science Department, Boston College, Chestnut Hill, Massachusetts, United States of America, **2** NeuroTechnology Center, Columbia University, New York, New York, United States of America

☉ These authors contributed equally to this work.
* weidf@bc.edu

## Abstract

Vesicles are critical components of neurons that package neurotransmitters and neuropeptides for their release, in order to communicate with other neurons and cells. However, due to their small size, the reconstruction of the full vesicle endowment across an entire neuronal morphology remains challenging. To achieve this, we have used, as a tool to identify and visualize vesicles, Volume Electron Microscopy (vEM), a method that has the nanoscale resolution to detect individual vesicle boundaries, content, and 3D locations. However, the large volume of vEM datasets poses a challenge in the segmentation, classification, and spatial analysis of tens of thousands of vesicles and their target cell in 3D. Here we report the development of VesiclePy, an integrated pipeline for automated segmentation, classification, proofreading, and spatial analysis of vesicles, relative to neuron masks in large-volume electron microscopy data. Our package integrates the efficiency of deep learning and the accuracy of human proofreading and provides a streamlined package in chunked processing and accurate indexing, localization, and visualization of single vesicle resolution in large vEM data. We demonstrate the viability of VesiclePy using high-pressure frozen serial EM data of *Hydra vulgaris* and quantify the performance of the package using ground truth manual annotations. We show that VesiclePy can process a multiterabyte serial EM dataset, efficiently annotate 53,851 vesicles from 20 complete neurons, and classify vesicles into 5 types. Each vesicle has a unique ID and 3D location for further spatial analysis in relation to neuron or non-neuronal targets nearby. Finally, by combining vesicle data and morphological information of each neuron, we can quantitatively cluster neurons into subtypes. VesiclePy is available at https://github.com/PytorchConnectomics/VesiclePy under an MIT license.

**Data availability statement:** VesiclePy is freely available under the MIT license on GitHub (https://github.com/PytorchConnectomics/VesiclePy) with an archive on Zenodo (https://doi.org/10.5281/zenodo.16644943). A sample of testing data can be found in the GitHub repository (https://github.com/PytorchConnectomics/VesiclePy/tree/main/ves_seg/sample).

**Funding:** The work in this publication was supported by the National Science Foundation (NSF; https://www.nsf.gov/) and United States Department of Defense (DOD; https://www.defense.gov/). R.Y. received support from CRCNS 1822550, NSF-2203119, and ONR N000142012828. D.W. received support from NSF-2239688. The funders had no role in study design, data collection and analysis, decision to publish, or preparation of the manuscript.

**Competing interests:** The authors have declared that no competing interests exist.

## Introduction

Neurons use vesicles to store neurotransmitters and neuropeptides and release them at the membrane to communicate with neighboring and distant cells. Understanding vesicle morphology and spatial distribution helps understand the content, mechanism, and target of neuronal communication. Given their small size and transient nature, electron microscopy is well-suited for capturing sub-cellular resolution vesicle membranes and structures. However, most analyses are limited to synapses and small vesicles using manual [1,2], or automated [3,4] segmentation, and large vesicles containing neuropeptides and their release site outside synapses have not been systematically characterized in 3D in full neurons. Here we present VesiclePy, a Python pipeline that provides a streamlined process for automated segmentation, classification, proofreading, spatial analysis, and visualization of vesicles, as well as the classification of neurons according to vesicle composition and neuron morphology. In this example dataset, we obtained 1,829 sections of serial electron microscopy (SEM) images and 20 neuron segmentations in half of the endoderm [5] from a *Hydra*. *Hydra* has a simple nervous system of a few hundred neurons, and a sparsely distributed nerve net with few neuronal contacts (Fig 1A). *Hydra*'s endodermal neurons contain a great number of large vesicles, including dense core vesicles (DCV), clear vesicles (CV, Fig 1B), dense core vesicles with halo (DCVH, Fig 1C), each of which may contain neuropeptides and neurotransmitters. There are also small clear vesicles (SCV) and small dense vesicles (SDV), mostly found in soma, and may not participate in synaptic release. The location of vesicles can inform us about their potential target, in this case, usually not near another neuron, suggesting asynaptic paracrine release. Each neuron contains all types of vesicles, however, the proportion of each vesicle type may differ across neuron types. To our knowledge, VesiclePy represents the first end-to-end framework integrating vesicle segmentation, classification, spatial analysis, and neuron-level clustering within a single pipeline, bridging the gap from pixel-level segmentation to network-level biological insights such as the unsupervised neuron typing demonstrated in this study.

## Design and implementation

We designed VesiclePy as a unified framework that processes large and small vesicles in parallel, facilitating the transition from raw volumetric data to high-level morphological clustering (Fig 2). While specialized tools like SynapseNet [4] and CryoVesNet [3] are highly effective for reconstructing vesicle clusters within specialized synaptic active zones, they are not intended for a cell-wide vesicle census. VesiclePy addresses this by providing a framework for broad vesicle analysis across entire cellular volumes. In this study, we utilize this infrastructure to map asynaptic release sites within a neuronal volume, but the pipeline is generalizable to any cellular context where the comprehensive distribution and morphological diversity of vesicles are of interest. By prioritizing the full organellar endowment, our approach enables researchers to define cell identities based on their total composition rather than localized connectivity alone.

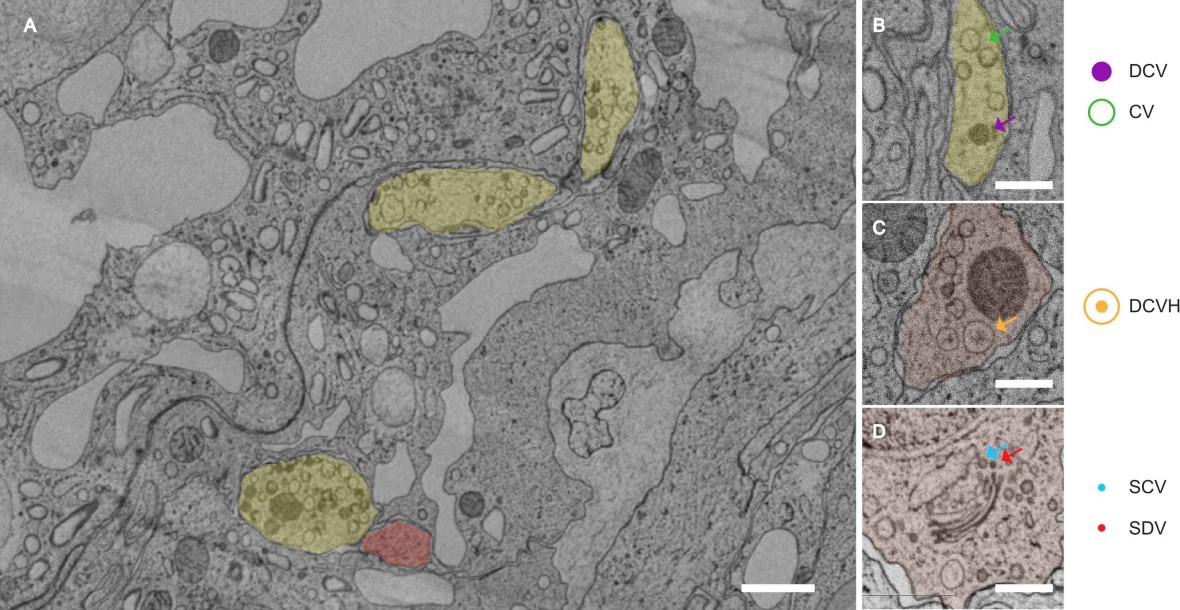

**Fig 1. EM image, neuron segmentation and five types of vesicles in *Hydra vulgaris*. A)** Representative EM section, with yellow segmentation over one neuron and red over another neuron, Scale bar 1 μm **(B)** A closeup EM image of dense core vesicles (DCV, purple arrow) and clear vesicles (CV, green arrow) in a neuron (yellow). **(C)** Two dense core vesicles with halo (DCVH, orange arrow) in a neuron (red). **(D)** Small clear vesicles (SCV, blue arrow) and small dense vesicles (SDV, red arrow) in the soma near the Golgi apparatus. **(B)**-**(D)** Scale bar 500 nm.

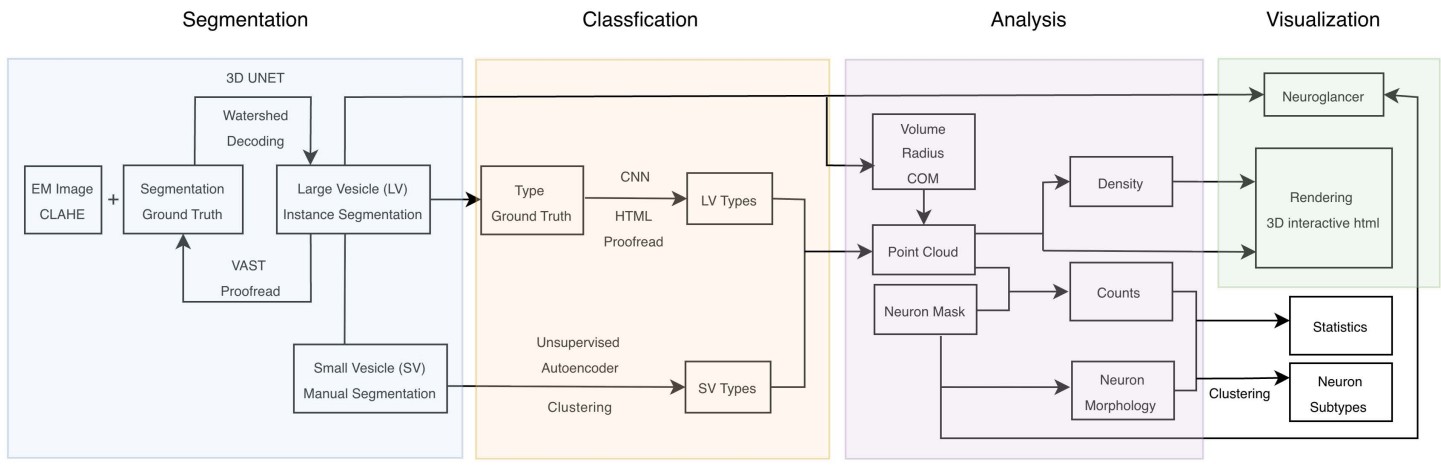

**Fig 2. Pipeline overview.** Flowchart of the computational pipeline in 5 modules.

Although demonstrated here on high-pressure frozen *Hydra vulgaris* serial SEM data, the modular design of VesiclePy does not rely on organism-specific priors. The segmentation backbone operates on voxel-level contrast and spatial continuity and can be retrained on other vEM datasets given appropriate ground-truth annotations. In principle, the pipeline can be applied to other large-scale volume electron microscopy datasets, including connectomics resources such as FlyWire

[6], MICrONS [7], or the H01 datasets [8], provided that appropriate vesicle annotations are available for model training or fine-tuning.

Nevertheless, performance may be affected by domain shift arising from differences in fixation protocol, imaging modality, voxel anisotropy, or signal-to-noise characteristics. In such cases, fine-tuning using a small curated subset of labeled volumes is recommended. The iterative human-in-the-loop correction cycle implemented in VesiclePy is designed to mitigate such cross-dataset variability efficiently.

Our pipeline consists of modular processing centers for segmentation, classification, analysis, visualization, and additional statistical and clustering analysis. A deep learning model which is iteratively trained with ground-truth data is used to segment large vesicles, and for difficult segmentation of small vesicles, a manual segmentation tool is provided. Vesicle subtypes are classified using a supervised approach for large organelles and a Variational Autoencoder (VAE) for the unsupervised identification of small vesicle populations. Beyond simple detection, the analysis module extracts 3D coordinates and morphological descriptors, which are leveraged to calculate local densities and spatial proximity to biological regions of interest. These metrics support multi-scale visualizations and provide the integrated feature set required to cluster neurons into subtypes based on their combined organellar and anatomical profiles (Fig 2).

**Large vesicle instance segmentation**

Vesicle detection and segmentation of vesicles individually in 3D is the foundation for any downstream analysis on vesicle morphology and locations. However, every neuron has thousands of sparsely distributed vesicles, most of which span multiple z-slices. Human annotation would be extremely labor-intensive and prone to errors. Thus, we used an iterative, semi-automated 3D-UNet [9] convolutional neural network, specifically the architecture described in the NucMM challenge [10], that learns from user-annotated volumes and provides multichannel 3D semantic segmentation predictions. Furthermore, we used CLAHE [11] as a preprocessing step to increase image contrast, and then applied multichannel watershed to achieve instance segmentation of individual vesicles. These predictions are further corrected by humans and learned by the model to produce better predictions. Additionally, digital enhancement of the image and sampling of erroneous regions further improved the prediction result.

**Iterative training and prediction.** Development followed a cyclical process of training, prediction, human correction, and sampling of erroneous regions to improve the model's performance. Firstly, we annotated multiple ground truth volumes and partitioned them into training, validation, and testing subsets using an approximate 80/10/10 split. Upon proofreading of the prediction result, we identified failure patterns and selected additional areas exemplifying targeted failure patterns that required improvement. We then generated predictions on these volumes and corrected any errors so that they could be used as ground truth in the next cycle of training and prediction. Here, we use the term training cycle to denote one full round of annotation, model training, prediction, and human correction. Training iterations reported in Table 1 refer to individual gradient update steps performed during model optimization. For example, after two training cycles using 26 volumes, 340,092,727 voxels of training data (Fig 3B-3C), we found poor qualitative performance in

**Table 1. Segmentation model training and performance.**

| Training Iterations | RAND | Precision | Recall | F1 Score | Example Result |
|---|---|---|---|---|---|
| 100,000 | 0.3446 | 0.9914 | 0.4982 | 0.6631 | Fig 3B |
| 300,000 | 0.16242 | 0.9848 | 0.7340 | 0.8412 | Fig 3C |
| 1,000,000 | 0.1517 | 0.9842 | 0.7492 | 0.8508 | Fig 3D |

Evaluation of model performance across checkpoints using RAND, precision, recall, and F1 scores. Fig 3A represents the raw EM input after CLAHE preprocessing but prior to model prediction. Metrics for 1,000,000 iterations correspond to the final optimized model used for dataset-wide segmentation.

PLOS Computational Biology

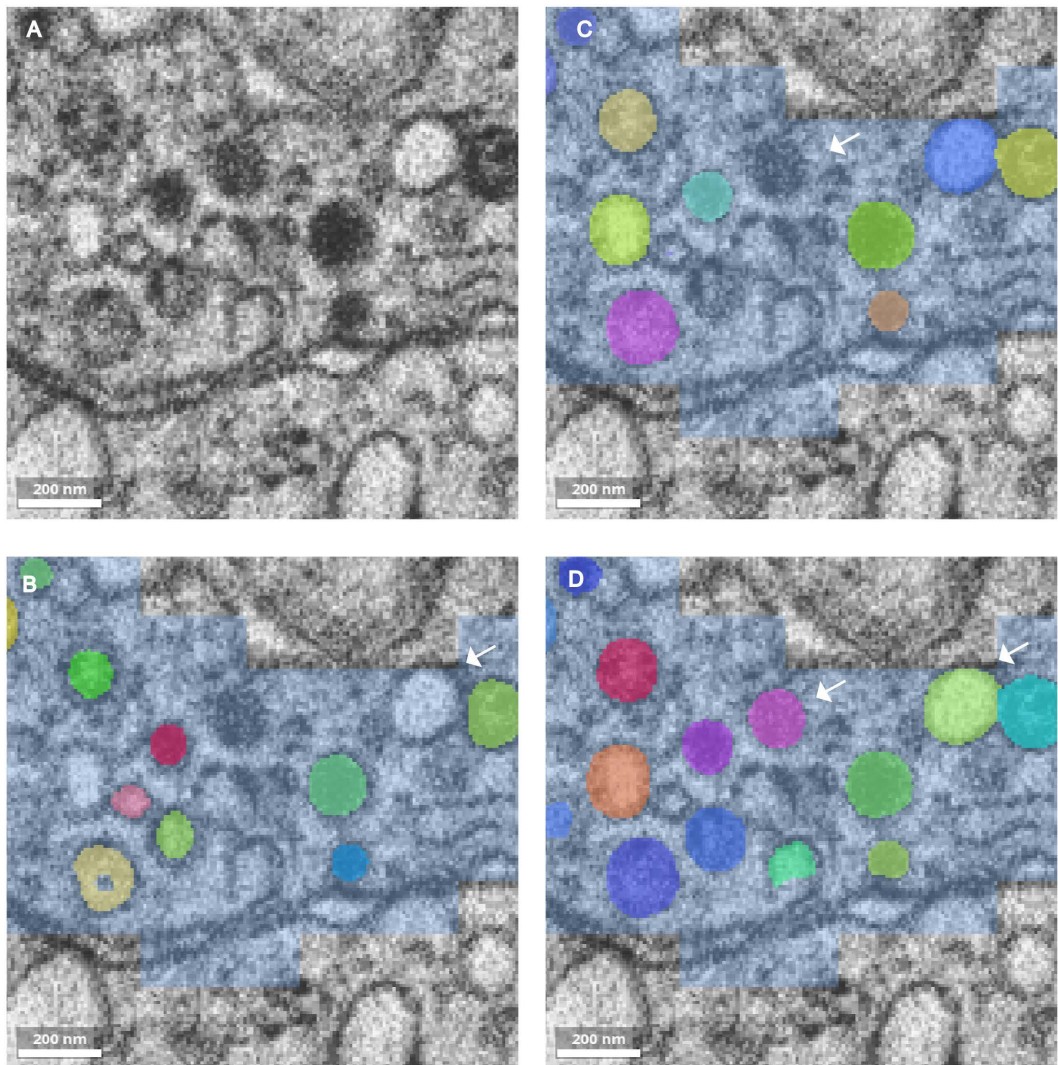

**Fig 3. Example results at training checkpoints and segmentation failure modes. (A)** EM image after applying CLAHE. **(B)** Prediction after 100,000 iterations; the white arrow indicates a false negative, while the yellow vesicle illustrates a failure case with internal "holes" due to early-stage training noise. **(C)** Result after 300,000 iterations; white arrow marks a false negative dense core vesicle. **(D)** Result after 1,000,000 iterations; white arrows mark improved results. Scale bar: 200 nm.

regions containing a high concentration of dense core vesicles and dense core vesicles with a halo. Using connected components, we find 33 different small subvolumes, 354,022,199 voxels that have a concentration of these dense vesicles, then manually correct them. These subvolumes were used as training data in the third training cycle (Fig 3D).

**Instance segmentation.** To achieve precise instance segmentation of vesicles, we predicted a three-channel volume comprising binary segmentation, contour, and distance transform maps. While the 3D U-Net [9,10] processes volumetric context and outputs the distance transform as a 3D target, the contour map is generated slice-wise (2D) to capture precise in-plane boundaries.

We collapsed this multi-channel prediction into single-channel labeled instances using a 3D seeded watershed algorithm [12]. Crucially, seeds are not generated via simple local maxima detection; instead, they are generated via a strict

threshold intersection across all three channels (high binary, low contour, high distance). Following connected-components labeling and small-seed removal, these regions serve as the starting points for flooding.

To mitigate contradictions between channels—such as the contour map predicting a boundary where the binary mask predicts foreground—the algorithm applies a strict hierarchy. The contour channel acts primarily as a seed suppressor; it does not automatically exclude voxels from the foreground flooding mask. The foreground mask is defined strictly by the binary and distance channels, while the actual watershed topography is driven by the inverted binary mask. Failure cases, such as vesicles with holes (Fig 3B) or partial reconstructions (Fig 3D), typically arise from noisy binary predictions. Our iterative pipeline addresses these topologies by sampling the erroneous subvolumes into subsequent training cycles. To assess model performance, we utilize the Foreground-Adjusted Rand Index (RAND), Precision, Recall, and the F1 score. The RAND index (lower is better) measures topological similarity between predicted and ground-truth labels. Precision and Recall (higher is better) quantify over-segmentation and under-segmentation, respectively. Finally, we include the F1 score as a balanced metric to evaluate the harmonic mean of precision and recall across the dataset.

## Supervised vesicle type classification

Morphology of vesicles is useful in determining their type, content, and state in a dynamic biological process. In the example dataset, we could identify three morphologically distinct large vesicle types as previously described: CV (Fig 4A), DCV (Fig 4B), and DCVH (Fig 4C). Each vesicle spans multiple z-sections, sometimes with varied appearances on different sections. We used a 3D convolutional neural network [13] to automate the classification process. We also developed an HTML-based proofreading tool that displayed the results of vesicles of the same kind on one page, which allowed for efficient human proofreading and correction. The network processes 31×31×5 voxel stacks using a series of convolutional layers with batch normalization and ReLU activations, followed by max pooling and fully connected layers to produce categorical predictions. Despite its architectural simplicity, the network captured essential morphological features relevant to vesicle classification. To mitigate class imbalance and improve generalization, we constructed a balanced training dataset by aggregating vesicles from all available manually proofread cells and sampling 1916 of each CV, DCV, and DCVH examples. When trained on this dataset and evaluated on a held-out collection of previously labeled data, the model achieved a validation accuracy of 83.6%. The evaluation result (Table 2) showed that the network was able to distinguish between vesicles of very distinct features (DCV and CV). However, vesicles with mixed morphology (DCVH) had a lower performance.

To ensure the robustness of the pipeline, we implemented an iterative workflow that integrates classification predictions with visualization and manual proofreading in HTML format. Input data consisted of 3D vesicle image stacks and their corresponding bounding box files, organized into neuron-specific subdirectories. The pipeline first applies the trained network to classify each vesicle and outputs its prediction along with a visualization image. These results are organized into category-specific subfolders (CV, DCV, DCVH), providing users with an overview of the classification result that can be proofread and corrected by simply clicking.

## Unsupervised small vesicle type classification

Small vesicles (< 80 nm) are manually annotated to ensure accurate identification (Fig 5A). Small vesicles are operationally defined as vesicles with a diameter of less than 80 nm; they are much smaller than the large vesicles and difficult to classify manually by morphology. Instead of training a supervised classification model to predict vesicle types, as we did in the previous section, here we performed unsupervised dimensionality reduction to determine whether distributional analysis (made feasible on a lower number of dimensions) can be used to naturally reveal clusters of vesicle types.

To standardize the inputs for the Variational Autoencoder (VAE), we extracted $1 \times 11 \times 11$ pixel 2D image patches from the 3D vesicle volumes. Although small vesicles may span 2–3 z-slices, the pipeline explicitly calculates the 3D bounding box midpoint for each vesicle and extracts exactly one 2D slice at this geometric center. No maximum intensity projection

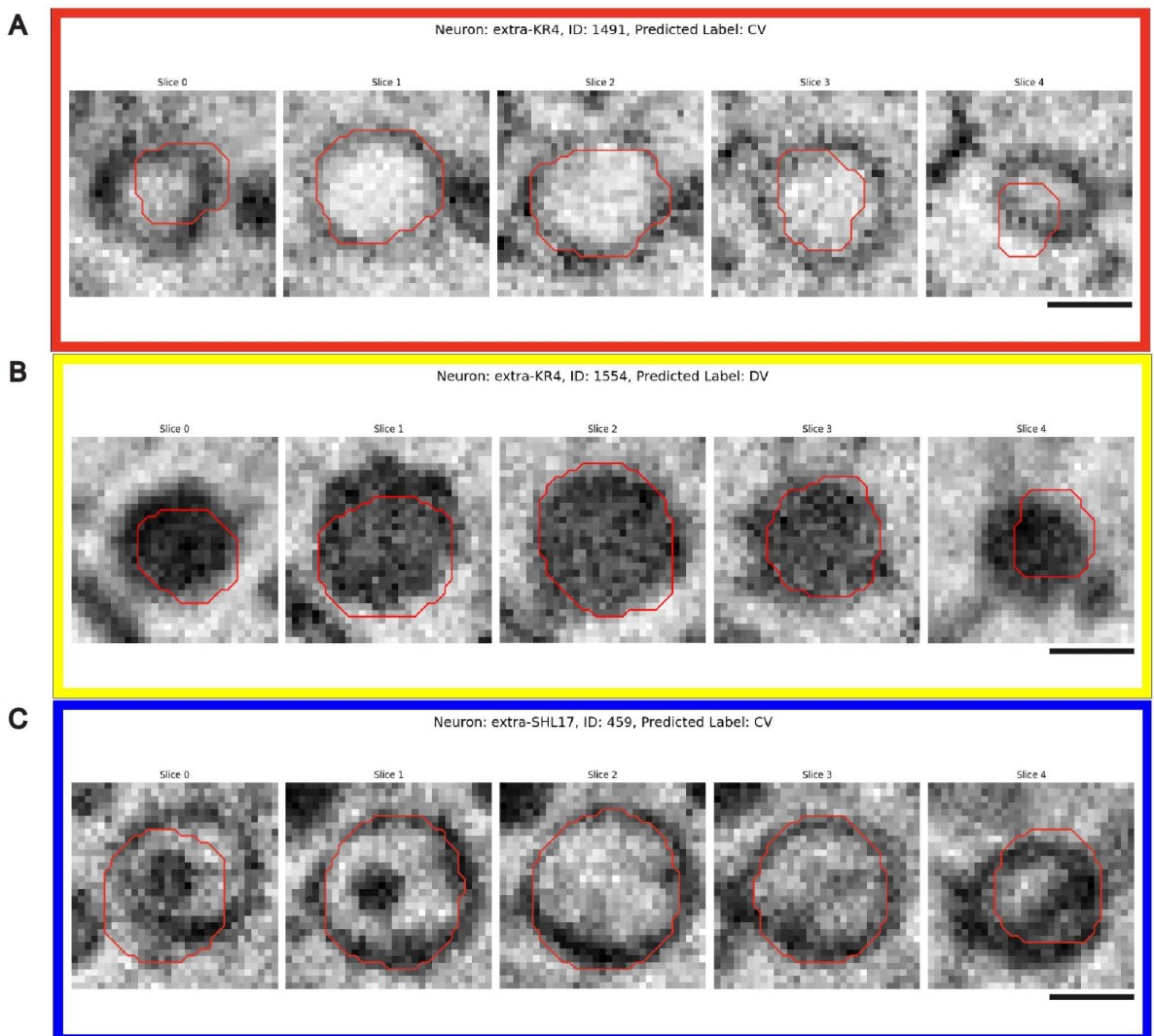

**Fig 4. Vesicle type classification proofreading interface.** The interface enables human-in-the-loop verification of automated vesicle classifications. For each candidate **(A)** Clear vesicle (CV), **(B)** Dense core vesicle (DCV), or **(C)** Dense core vesicle with a halo (DCVH), the interface presents five consecutive Z-slices (Slices 0–4) overlaid with the contour of the 3D U-Net prediction in red, which allows users to rapidly assess volumetric morphology and label accuracy. Scale bar 200 nm.

**Table 2. Classification model performance on different vesicle types.**

| Type | RAND | Precision | Recall | F1 Score |
|---|---|---|---|---|
| CV | 0.3125 | 0.6931 | 0.6820 | 0.6875 |
| DCV | 0.0991 | 0.9099 | 0.8921 | 0.9009 |
| DCVH | 0.8655 | 0.0941 | 0.2353 | 0.1344 |
| Total | 0.4162 | 0.5657 | 0.6031 | 0.5838 |

Evaluation of the model performance for each type of large vesicle and in total, measured by RAND, precision, recall, and F1 scores.

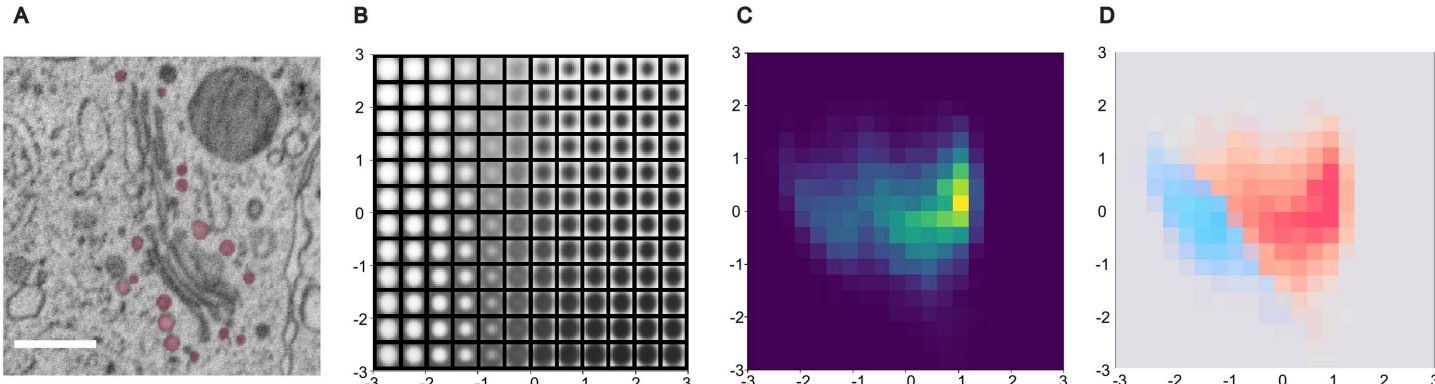

**Fig 5. Unsupervised small vesicle type classification. (A)** Manual semantic segmentation of small vesicles. Scale bar 500 nm. **(B)** Variational autoencoder latent manifold reconstructions across the 2D latent space. **(C)** Embedding distribution heatmap utilizing a Viridis color map, where yellow indicates the highest density of vesicle instances. **(D)** HDBSCAN clustering of vesicle types overlaid on a density map using a diverging blue-to-red color map to visualize cluster separation.

(MIP) is used. This center-slice extraction inherently avoids 'edge-position' slices (the top or bottom caps of the vesicle), which typically suffer from partial-volume effects and lower contrast. Given the high in-plane resolution (4 nm) relative to the section thickness (30 nm), this 2D geometric center is highly representative of the vesicle's morphology.

To account for potential translations and rotations, we utilized a convolutional VAE with in-built translation and rotation invariance provided by the pyroVED [14] library. The VAE latent dimensionality is fixed at 2. Consequently, each 11 × 11 image patch is compressed into a two-dimensional embedding, where the plotted x- and y-axes represent the raw VAE latent means ($\mu_1$ and $\mu_2$) generated by the encoder; no Principal Component Analysis (PCA) or further dimensionality reduction is applied. The resulting distribution is visualized as a heatmap using a Viridis color map, where yellow indicates the highest density of morphologically similar vesicle instances (Fig 5C). Finally, we applied the HDBSCAN [15] algorithm directly on these two latent coordinates to cluster the data into two distinct categories, visualized through a diverging blue-to-red density map (Fig 5D).

## Morphology and spatial analysis

Unlike small vesicles, large and dense core vesicles are often released in a paracrine manner outside of synapses [5,16]. Analyzing their spatial distribution and potential targets requires examining large regions spanning the entire neuron and its neighbors. To manage this computationally intensive task, we confined analysis to minimal bounding boxes per neuron, stored in high-resolution HDF5 format. Vesicle data within these regions were converted into point clouds that retain key morphology, spatial, and segmentation metadata for efficient downstream processing.

**Neuron mask usage.** To identify regions of possible vesicle release that target another neuron, we stitched adjacent neurons into unified bounding boxes using global coordinates, and identified regions "near" another neuron by computing a Euclidean Distance Transform [17] from the stitched pieces—based on a threshold determined by the average vesicle diameter plus 2 times standard deviation (Fig 6). To account for segmentation uncertainty, we added 1–3 voxel buffers around both source and adjacent neurons, then extracted surface patches within these margins. The voxel counts of these regions were scaled by resolution to compute surface area.

**Vesicle morphology.** The high resolution needed to compute vesicle morphology and the large number of vesicles in each neuron demand extensive computational resources. Therefore, we divided the task into three parts, each leveraging the appropriate file format. Firstly, we computed the morphology of each vesicle using the original segmentation masks.

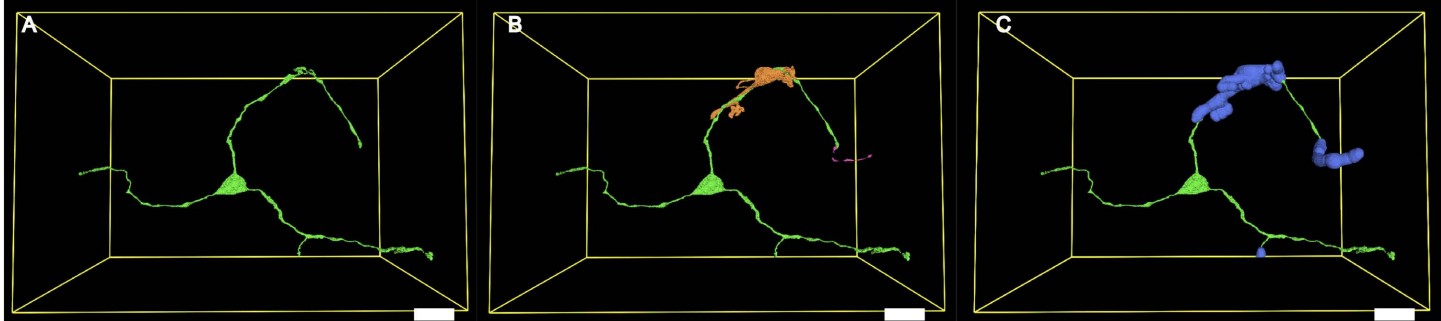

**Fig 6. Neighbouring neuron and potential vesicle release region. (A)** 3D rendering of a neuron (green) within its minimal bounding box (yellow). **(B)** The same neuron (green) with adjacent neuron fragments (orange) that fall within the bounding box. **(C)** Expanded adjacent neuron masks (blue) used to define the spatial region considered "near another neuron" within the target neuron volume. **(A)–(C)** Scale bar 10 μm.

Vesicle volumes were calculated by summing segmented voxels and scaling by resolution. Radii were estimated by collapsing each roughly spherical mask along its lowest-resolution axis and measuring the equivalent circular area. Secondly, vesicles were converted into point clouds storing their center of mass (COM), volume, radius, classification type, and ID. This format allowed for compact exports and efficient analysis using tools like Pandas [18] and Polars [19], which are poised to process column-based data. Finally, vesicle distribution densities were calculated instance-wise using kd-trees [20], undergoing a nearest neighbors query within 500 nm of each vesicle COM. These values were normalized across the dataset to visualize relative density gradients.

**Counting vesicles.** Finally, to determine the number of vesicles within a specific neuronal region of interest, we extracted values of a binary mask of the region from all of the stored COM coordinates of the vesicles within the neuron, thus avoiding direct use of the original high-resolution vesicle segmentation data. For example, we calculated the number of vesicles present in "near-neuron" regions by using distance thresholds to extract masks of these regions, then utilizing the point cloud metadata to keep track of vesicle type distributions. Similarly, we calculated the number of small vesicles present within soma regions to determine whether there was a greater density of small vesicles within the soma.

## Visualization

Effective visualization remains a persistent challenge in the analysis of large-scale volume electron microscopy (vEM) datasets [21], where the complexity and density of biological structures require tailored tools for rendering, inspection, and semantic interpretation. To support efficient human-in-the-loop data analysis and observational hypothesis generation, we implement a modular visualization pipeline built on four complementary platforms—Neuroglancer [22], PyVista [23], Plotly [24], and three.js [25]—each addressing specific bottlenecks in interoperability, scalability, and semantic clarity.

Neuroglancer (Fig 7A) anchors the workflow by allowing direct inspection of EM data and segmentation overlays for ground truth validation. Although its format limitations restrict customization (e.g., dynamic color encodings or metadata overlays), it anchors all downstream visualization through persistent vesicle ID links across tools. PyVista (Fig 7B) enables more flexible, high-fidelity mesh visualization, where features like vesicle subtype or spatial density can be encoded directly as colormaps. However, PyVista's performance degrades on full-volume datasets and requires substantial preprocessing, limiting its scalability for larger data exploration tasks. Plotly (Fig 7C), allows for browser-native 3D rendering of vesicle point clouds and mesh overlays, with hover-based metadata inspection and standalone HTML export. This allows for quick inspection of individual neurons or vesicle subsets, but struggles to scale to full datasets due to computational constraints. For dataset-wide visualization at scale, we develop a lightweight three.js-based viewer (Fig 7D) optimized

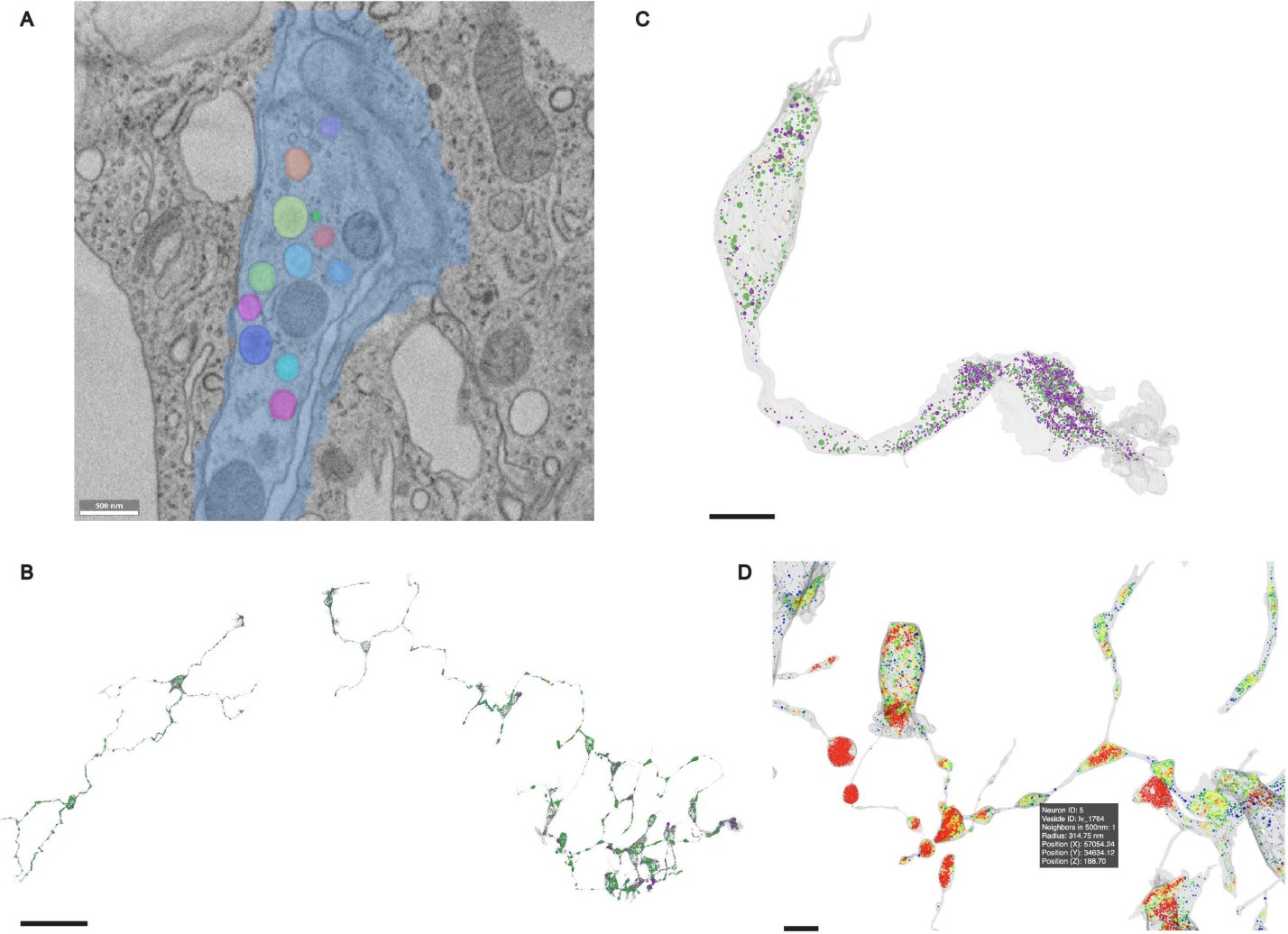

**Fig 7. A four-platform visualization pipeline for vEM data analysis.** A four-platform visualization pipeline for vEM data analysis. **(A)** Neuroglancer anchors ground truth validation via EM overlays and segmentation. Scale bar 500 nm. **(B)** PyVista supports high-fidelity local mesh rendering with customizable colormaps, here representing vesicle subtypes. Scale bar 50 μm. **(C)** Plotly enables interactive browser-based mesh and point cloud inspection, shown here using a subtype colormap. Scale bar 5 μm. **(D)** Three.js provides scalable full-dataset visualization with flexible metadata mapping, demonstrated here with a density-based colormap. Scale bar 5 μm.

for rendering tens of thousands of vesicles and neurons with subtype and density encodings. Though it lacks raw image overlays, it fully supports metadata and links back to Neuroglancer, preserving semantic disambiguation without sacrificing rendering performance.

## Neuron type cluster analysis

Neuron morphology and connectivity are common parameters for clustering of neuron types in connectomics studies [26,27]; however, when connectivity extends beyond physical synaptic connections, we propose the combination of morphology and vesicles as additional valid clustering parameters. In this example dataset, an initial categorization of 20 neurons into 5 distinct clusters was established from distinctive morphology features [5]. To further validate these clusters,

we utilized both morphology and vesicle information in an unsupervised classification method. Vesicle compositions of each neuron from the previous analysis, and several categories of morphology data (polarity, cilia, microvilli, handshake, volume, and length) for the neurons were extracted. Numerical features were logarithmically normalized from [0,1]. Subsequently, the Gower distance metric was calculated from this processed mixed-data feature set, producing a 20×20 pairwise dissimilarity matrix. Gower distance provides a unified similarity measure across these diverse feature types, enabling a holistic comparison relevant to overall neuron structure [28]. The vertical axis of the resulting dendrogram (Fig 8) represents the absolute Gower dissimilarity, which calculates the average difference across the morphological and organellar characters. In this representation, a height of 0.0 signifies identical feature profiles, while 1.0 represents the maximal theoretical divergence across all parameters. Using this Gower matrix, hierarchical clustering with the 'complete' linkage method was performed [29]. Cutting the resulting dendrogram at a distance threshold of 0.4 yielded 4 distinct neuron clusters (Fig 8) that corresponded with 4 clusters established through qualitative morphology classification, and 1 cluster (Neuron 10) merged.

## Results

The example dataset consists of a scanning electron microscopy (SEM) volume acquired from high-pressure frozen *Hydra vulgaris*. Serial sections were collected at 30 nm thickness, yielding an effective voxel resolution of 4 × 4 × 30 nm. Imaging was performed on a Zeiss SIGMA scanning electron microscope using Atlas 5 (Zeiss Microscopy) mapping software, with acquisition parameters of 1000 ns dwell time, 4 nm in-plane pixel size, and back-scattered electron detection. The resulting EM sections were subsequently stitched and aligned using the open-source Python package feabas [30]. The pipeline segmented a total of 53,851 vesicle instances, including 42,235 large vesicles and 11,616 small vesicles in 20 complete segmented neurons. We classified all large vesicles into these three categories, yielding 24,998 CV, 15,188 DCV, and 2049 DCVH. For small vesicles, we classified 2305 SCV and 9311 SDV. Next, we extracted the volume and diameter of each vesicle and computed the overlap between the vesicle coordinates with the "near" and "far" from another neuron region, and quantified the percentage of vesicles "near" another neuron is significantly less than "far" across different neuronal types and vesicle types. We also created 2D rendering and 3D interactive HTML for visualization and inspection

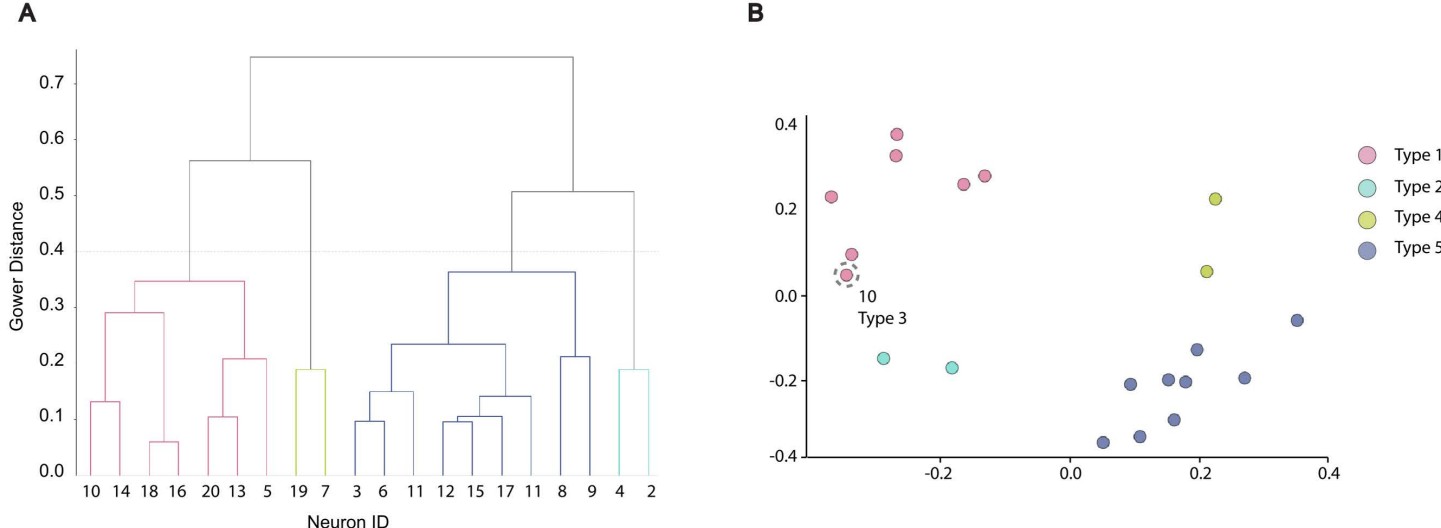

**Fig 8. Neuron type cluster analysis.** A dendrogram of neuron clustering based on combined morphological and vesicle features. Pairwise dissimilarity is measured by absolute Gower dissimilarity calculated across the mixed-data feature set, with a threshold of 0.4 yielding 4 distinct clusters.

of all vesicle types, their location, and density in relation to the neuron mask. Finally, using vesicle count by type for each neuron and additional morphological measurements and features, we were able to cluster neurons into 4 subtypes that aligned with our initial qualitative classification based on morphology.

Dataset-wide processing was executed on a high-performance computing cluster utilizing two dedicated compute nodes, each equipped with 64 CPU cores, 512 GB of RAM, and four NVIDIA A10 GPUs (24GB VRAM each). The most computationally intensive component of the VesiclePy workflow is the training of the 3D U-Net segmentation model. In our experiments, model training required approximately 10 hours using four NVIDIA A10 GPUs. Once the segmentation model has been trained, the remaining stages of the pipeline—including vesicle classification, spatial analysis, and visualization—are substantially lighter computationally. In particular, many downstream analyses operate on compact point cloud representations of vesicles rather than the original volumetric data, enabling efficient processing of tens of thousands of vesicles with minimal memory overhead. Consequently, runtime depends more on the analysis performed than on dataset size alone. While the full aligned serial EM volume spans approximately 45.9 trillion voxels (~45.9 TB at `uint8`), our analysis focused on the cumulative image data contained within the 20 individual neuronal bounding boxes, which totaled 2.75 TB. To facilitate this scale of analysis, workload distribution was managed via Slurm-based job sharding, utilizing a chunked HDF5 storage backend with Gzip compression to maintain an out-of-core parallelization strategy.

## Availability and future directions

VesiclePy is an open-source software package available at https://github.com/PytorchConnectomics/VesiclePy under the MIT license. The repository includes source code, installation instructions, documentation, and example workflows for segmentation, classification, and spatial analysis of vesicles in volumetric electron microscopy data.

The modular design of VesiclePy enables straightforward adaptation to new datasets and biological contexts. While demonstrated on high-pressure frozen *Hydra vulgaris* serial electron microscopy data, the segmentation and analysis components can be retrained or fine-tuned on other volume electron microscopy datasets with appropriate annotations.

Future work will focus on improving robustness across imaging domains, including handling variability in contrast, resolution, and noise characteristics. This includes incorporating domain adaptation strategies and reducing the annotation burden required for retraining. Additionally, we plan to expand support for fully automated small vesicle segmentation and classification, reducing reliance on manual annotation and enabling more scalable analysis of dense vesicle populations.

We also aim to enhance the integration between VesiclePy and existing visualization and data platforms, including tighter coupling with Neuroglancer and web-based visualization frameworks for large-scale interactive exploration. Further development will explore richer multimodal representations that combine vesicle morphology, spatial organization, and neuron-level features to support more comprehensive cell-type classification and functional inference.

Finally, as vesicle composition provides an additional axis for characterizing neuronal identity beyond connectivity alone, future extensions of VesiclePy may enable integration with complementary modalities such as molecular or transcriptomic data to facilitate more complete multi-scale models of neural systems. In parallel, larger annotated neuron cohorts will allow future work to move beyond the proof-of-concept clustering analysis presented here by incorporating quantitative cluster-validation metrics such as silhouette scores, cophenetic correlation, and bootstrap stability.

## Author contributions

**Conceptualization:** Rafael Yuste, Donglai Wei.

**Data curation:** Paige Nurkin, Shulin Zhang.

**Formal analysis:** Jason Ken Adhinarta, Yutian Fan, Adam Gohain, Michael Lin, Paige Nurkin, Richard Ren, Micaela Roth, Shulin Zhang.

**Funding acquisition:** Rafael Yuste, Donglai Wei.

**Investigation:** Rafael Yuste, Donglai Wei.

**Project administration:** Shulin Zhang, Rafael Yuste, Donglai Wei.

**Software:** Jason Ken Adhinarta, Yutian Fan, Adam Gohain, Michael Lin, Richard Ren, Micaela Roth, Ayal Yakobe.

**Supervision:** Rafael Yuste, Donglai Wei.

**Validation:** Rafael Yuste, Donglai Wei.

**Writing – original draft:** Jason Ken Adhinarta, Yutian Fan, Adam Gohain, Michael Lin, Richard Ren, Micaela Roth, Shulin Zhang.

**Writing – review & editing:** Jason Ken Adhinarta, Yutian Fan, Adam Gohain, Michael Lin, Richard Ren, Micaela Roth, Shulin Zhang, Rafael Yuste, Donglai Wei.

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
