## [Decision Letter · Decision Letter 0]

30 Dec 2025

PCOMPBIOL-D-25-01757

VesiclePy: A Machine Learning Vesicle Analysis Toolbox for Volume Electron Microscopy

PLOS Computational Biology

Dear Dr. Wei,

Thank you for submitting your manuscript to PLOS Computational Biology. After careful consideration, we feel that it has merit but does not fully meet PLOS Computational Biology's publication criteria as it currently stands. Therefore, we invite you to submit a revised version of the manuscript that addresses the points raised during the review process.

We look forward to receiving your revised manuscript.

Kind regards,

Drew Linsley

Guest Editor

PLOS Computational Biology

Thomas Serre

Section Editor

PLOS Computational Biology

**Journal Requirements:**

**Reviewers' comments:**

Reviewer's Responses to Questions

**Comments to the Authors:**

Reviewer #1: The manuscript presents VesiclePy, an integrated pipeline for processing large vEM datasets to segment, classify, curate, analyze spatial relationships, and visualize vesicles in neurons. The workflow appears comprehensive, with convincing reconstruction and visualization outputs. It should be of interest to readers in the field. However, there are several issues related to technical details, figures, and methodological limitations that require optimization and clarification.

1.Dataset description.

The manuscript lacks explicit, self-contained descriptions of the dataset. Relying on prior work without providing related details may confuse readers about the data used here. Please describe key dataset information directly in the manuscript, including resolution and imaging modality. For example, in the cited prior work [5], the sample was “Approximately half of the sample (0.05 mm thickness) was then cut along the oral-aboral plane into 1,829 thin sections (30 nm thick), collected and imaged with a scanning electron microscope at 4 nm resolution, and stitched and aligned to form image stacks.” Please describe these specifics (resolution, sectioning, imaging modality, alignment/stitched stacks) within the present study.

2.Novelty and generalization.

The method has been evaluated only on Hydra vulgaris datasets and has not been extended to other datasets. Clarify whether the method is applicable to other vEM datasets (e.g., FlyWire, MICrONS, H01) or to other vesicle types, or whether it is limited to Hydra vulgaris. Also address whether this workflow represents the first fully end-to-end analysis toolbox in the field and articulate the main innovations. The discussion, limitations, and contributions could be strengthened with explicit statements about generalizability and scope, or with preliminary results on additional datasets if feasible.

3.Pipeline efficiency.

The claim that “VesiclePy can process a multiterabyte serial EM dataset” is compelling, but the manuscript does not provide concrete demonstrations of data size, throughput, or practical resource/time requirements. Please include: dataset sizes used in experiments, processing times, hardware specifications (CPU/GPU, RAM, storage), and any parallelization strategies. A discussion of bottlenecks and scalability would be helpful.

4.Instance segmentation details.

For the 3D U-Net outputs (distance transform and edge maps), clarify whether these are computed in 2D or 3D, and discuss how the choice impacts seed generation. Also elaborate on the statement “we collapsed the three-channel prediction into a single-channel prediction of individual vesicles.” Provide sufficient detail to explain how the reduction is performed and how potential contradictions between channels are mitigated. It would help to illustrate potential failure cases, such as the yellow vesicle with holes in Fig. 3B and the partially reconstructed green vesicles in Fig. 3D, and discuss how the method addresses such issues.

5.Unsupervised classification details.

In the Unsupervised Small Vesicle Type Classification section, vesicles defined as diameter < 80 nm may span 2–3 z-slices, yet the self-supervised classification uses 2D patches (11×11). Clarify whether edge-position patches could affect classification, and whether extending the receptive field in the z-dimension or using sequential 3D patches could improve results. Regarding the variational autoencoder features, what is the dimensionality of the latent embedding? The manuscript states “Each 11×11 image patch is compressed into an embedding with two latent dimensions.” Specify the exact technique (e.g., PCA or a VAE encoder) and rationale. For Fig. 5, indicate what the x- and y-axes represent (e.g., PCA projections) and label them accordingly if PCA is used (PCA Projection 1, PCA Projection 2).

6.Figure annotations.

Several figures would benefit from additional annotations. For Fig. 4, include a scale bar and clarify what the red contour lines represent, especially if they do not perfectly align with vesicle boundaries. For Fig. 8, define the vertical axis precisely. Ensure all captions and overlays consistently explain color codes, contours, and annotations to avoid confusion.

7.Minor errors.

Clarify potential citation numbering (e.g., “NucMM challenge (19)” may refer to [19]), and ensure figure references follow journal conventions (e.g., “Fig. x” rather than “Fig x”). A careful language and formatting check is recommended.

Reviewer #2: This work presents a pipeline for segmenting, classifying, visualizing different types of vesicles from Hydra's endodermal neurons. A practical solution is proposed to deal with different sizes of vesicles using both supervised trained deep learning models and a manual annotation tool. A merit of this work refers to its application of the proposed tool for morphological analysis which clearly visualizes the potential contact regions among neighboring neurons.

I have the following comments for the authors to check:

1. Some expressions in the manuscript may be improved. For example, from line 28-30: "Segmentation consists of an iterative deep learning model to automatically segment large vesicles from ground truth data, and manual segmentation of small vesicles.". I guess a more direct expression may be like this: "A deep learning model which is iteratively trained with ground-truth data is used to segment large vesicles, and for difficult segmentation of small vesicles, a manual segmentation tool is provided."

2. Colormap representations in figure 5 and 6 should be clarified.

3. The evaluation metrics should be clarified, for example, the definition of RAND, precision and recall, and how these metrics assess the performance of an algorithm? The larger the bettor or in the other way around. F1 score should be included for a balanced evaluation.

4. The authors may provide more details of the training, validation and testing datasets, for example, how many vesicles of each category are contained in each subset for each training step?

5. The results shown in Table 1 seems disagree to the content of figure 3. For example, in the caption of figure 3(B), it shows the results obtained from the experiments of 100,000 training epochs.

6. Please carefully check the definitions of "training iterations (steps)" and "training epochs". Regarding "one epoch", we usually mean using all training data to iteratively train the model for one time. 1,000,000 epochs seems an extremely long training process.

7. The title of figure 4B should be DCV not DV. And the items (A) (B) (C) are also missing in this figure.

8. How is the generalizability of the proposed pipeline, when used in other cases, like for the data obtained from different types of SEM or different types of neurons? Suppose that the domain shift may result in a decreasing performance, re-training the model or re-labeling the new data may be laboring.

**Have the authors made all data and (if applicable) computational code underlying the findings in their manuscript fully available?**

Reviewer #1: Yes

Reviewer #2: Yes

PLOS authors have the option to publish the peer review history of their article (what does this mean?). If published, this will include your full peer review and any attached files.

Reviewer #1: No

Reviewer #2: No

**Figure resubmission:**
---

## [Decision Letter · Decision Letter 1]

14 Apr 2026

Dear Professor Wei,

We are pleased to inform you that your manuscript 'VesiclePy: A Machine Learning Vesicle Analysis Toolbox for Volume Electron Microscopy' has been provisionally accepted for publication in PLOS Computational Biology.

Best regards,

Drew Linsley

Guest Editor

PLOS Computational Biology

Thomas Serre

Section Editor

PLOS Computational Biology

Please address the minor comments from R1 in your final version.

Reviewer's Responses to Questions

**Comments to the Authors:**

Reviewer #1: I thank the authors for their detailed, point-by-point Response to Reviewers and for the careful revisions implemented in the R1 version. The manuscript has been substantially improved in clarity, technical transparency, self-containment, and reproducibility.

One very minor methodological point remains regarding the neuron-type clustering analysis. The authors apply Gower distance + hierarchical clustering with complete linkage on a modest sample of n = 20 neurons and cut the dendrogram at an arbitrary threshold of 0.4. While the biological interpretation is reasonable and the result aligns with the prior qualitative classification, the small sample size and lack of any cluster-validation metrics (e.g., silhouette score, cophenetic correlation coefficient, or bootstrap stability) make the robustness of the obtained clusters difficult to assess quantitatively.

I do not consider this a major flaw, as clustering is presented as a demonstration rather than the central contribution. However, I recommend that the authors briefly acknowledge this limitation and frame it as a direction for future work. A concise sentence could be added to the Discussion or Availability and Future Directions section

**Have the authors made all data and (if applicable) computational code underlying the findings in their manuscript fully available?**

Reviewer #1: None

PLOS authors have the option to publish the peer review history of their article (what does this mean?). If published, this will include your full peer review and any attached files.

Reviewer #1: No